# Big Data in Criteria Selection and Identification in Managing Flood Disaster Events Based on Macro Domain PESTEL Analysis: Case Study of Malaysia Adaptation Index

**Mohammad Fikry Abdullah** [1,2,*] , **Zurina Zainol** [2] , **Siaw Yin Thian** [2] , **Noor Hisham Ab Ghani** [2] , **Azman Mat Jusoh** [2] , **Mohd Zaki Mat Amin** [2] and **Nur Aiza Mohamad** [2]

1  Leeds University Business School, University of Leeds, Leeds LS2 9JT, UK; zurina@nahrim.gov.my (Z.Z.); siawyin@nahrim.gov.my (S.Y.T.); noorhisham@nahrim.gov.my (N.H.A.G.); azman@nahrim.gov.my (A.M.J.); zaki@nahrim.gov.my (M.Z.M.A.); nuraiza@nahrim.gov.my (N.A.M.)
2  National Water Research Institute of Malaysia (NAHRIM), Seri Kembangan 43300, Malaysia
*  Correspondence: fikry.abdullah@gmail.com or fikry@nahrim.gov.my

**Abstract:** The impact of Big Data (BD) creates challenges in selecting relevant and significant data to be used as criteria to facilitate flood management plans. Studies on macro domain criteria expand the criteria selection, which is important for assessment in allowing a comprehensive understanding of the current situation, readiness, preparation, resources, and others for decision assessment and disaster events planning. This study aims to facilitate the criteria identification and selection from a macro domain perspective in improving flood management planning. The objectives of this study are (a) to explore and identify potential and possible criteria to be incorporated in the current flood management plan in the macro domain perspective; (b) to understand the type of flood measures and decision goals implemented to facilitate flood management planning decisions; and (c) to examine the possible structured mechanism for criteria selection based on the decision analysis technique. Based on a systematic literature review and thematic analysis using the PESTEL framework, the findings have identified and clustered domains and their criteria to be considered and applied in future flood management plans. The critical review on flood measures and decision goals would potentially equip stakeholders and policy makers for better decision making based on a disaster management plan. The decision analysis technique as a structured mechanism would significantly improve criteria identification and selection for comprehensive and collective decisions. The findings from this study could further improve Malaysia Adaptation Index (MAIN) criteria identification and selection, which could be the complementary and supporting reference in managing flood disaster management. A proposed framework from this study can be used as guidance in dealing with and optimising the criteria based on challenges and the current application of Big Data and criteria in managing disaster events.

**Keywords:** Big Data; PESTEL analysis; disaster management plan; macro domain criteria; flood criteria; decision analysis

## 1. Introduction

The impact of climate change on the country cannot be underestimated. As a result of global warming, water-related disasters, such as floods, will harm and impact the economy, society, and environment. Flood events are among the common disasters recorded due to climate change's impact. Changes in rainfall pattern and volume will affect the flood disaster magnitude and frequency. As part of the disaster management plan (DMP), specific flood mitigation and adaptation actions have been taken and implemented to reduce the risks and impact of climate change on flood events. The measures taken were initiatives to limit the activities that directly cause more harm. These mitigation and adaptation measures can be effectively impactful if stakeholders and policy makers understand the

degree of vulnerability and readiness in various significantly relevant domains for the flood management plan.

As part of the mitigation plan under climate change action, the National Water Research Institute of Malaysia (NAHRIM) took an initiative to minimise and mitigate the impact of climate change through the implementation of the Malaysian Adaptation Index (MAIN). MAIN was developed to examine and calculate the index adaptation through vulnerability and readiness sectors. The criteria from various sectors were identified, selected, and analysed to determine Malaysia's vulnerability and readiness status. The MAIN index can facilitate stakeholders and policy makers to strategise relevant and significant plans to improve state sustainability and resistance towards climate change impacts. In the case of flood events, cohesive and collective decisions based on disaster management plan (DMP) phases could be planned thoroughly based on the MAIN index value.

In doing so, stakeholders and policy makers must quantify the vulnerability and readiness from various domains and their relevant sectors based on selected criteria. The criteria were identified and selected thoroughly by a group of experts (technical and non-technical), stakeholders, policy makers, and academicians. Based on the calculated scores and index of vulnerability and readiness, it shows the level of vulnerability and readiness which can aid decisions in the flood management plan. Thus, comprehensive and collective data, information and knowledge expertise need to be gathered to support the criteria identification and selection process. Apart from that, a structured mechanism is required to achieve more effective and collective flood management planning.

As one of the emerging and discussed technologies, Big Data offers advantages in supporting the implementation of the flood management plan. Voluminous, veracity and varsity data are among the positive impacts gained through Big Data, allowing more analytical processes to be conducted. This scenario has created more opportunities for stakeholders and policy makers to identify and select data as indicators or criteria to aid decision making.

With the emergence of Big Data as the fundamental technology, MAIN was developed to enhance the data analysis process by using a voluminous amount of data from different domains. The implication of this scenario aids the decision-making process, where more data are made available as criteria for vulnerability and readiness sectors.

While the stakeholders and policy makers benefited from overwhelming data for criteria, concerns and challenges were raised in identifying and selecting the appropriate data (rank and priorities) to become the criteria for both sectors. Diverse preferences, conflict of interests, data selection, and data availability are among the issues that stakeholders and policy makers will encounter. Criteria for both sectors need to be identified and selected properly to represent the vulnerability and readiness status. Therefore, a structured mechanism called multi-criteria decision analysis (MCDA) is ideal to be incorporated into the process. In addition, exploring criteria from different domains could widen the potential criteria to be chosen in the future.

This study aims to facilitate the criteria identification and selection from a macro domain perspective in improving flood management planning. The objectives of this study are (a) to explore and identify potential and possible criteria to be incorporated in the current flood management plan; (b) to understand the type of flood measures and decision goals implemented to facilitate flood management planning decisions; and (c) to examine the possible structured mechanism for criteria selection based on the decision analysis technique.

This paper is structured based on the following sections: Section 2 discusses the background of this study; Section 3 describes the methodology used; and in Section 4, the criticism of the existing literature is provided. Section 5 provides a discussion and potential research, followed by the study's conclusions in Section 6.

## 2. Background of Study

### 2.1. Malaysia Adaptation Index

Malaysia's commitment to addressing the effects of climate change began in 2009 with the United Nations Framework Convention on Climate Change (UNFCCC) and the 15th Conference of Parties (COP) in Copenhagen. Since then, Malaysia has begun to stress the implementation of climate change adaptation through COP25 (2015), particularly in the water security, coast, food, and health sectors. Meanwhile, in COP26, Malaysia will focus on climate ambitions, financing, and carbon markets.

MAIN, a project initiated by NAHRIM, began in 2019 to respond to the importance of climate change adaptation in Malaysia for various settings in multiple sectors. Part of its initiatives is strengthening and supporting the Malaysia National Climate Change Policy prepared in 2009, integrating the responses and actions based on indicators, and increasing the resilience in climate change. The execution and plan align with the United Nations 2030 Sustainable Development Goal (SDG), especially SDG 13—Take immediate action to prevent climate change and its consequences. In line with the National Climate Change Roadmap and Adaptation plan, MAIN is expected to benefit various parties by enhancing and strengthening the national development plan based on climate resilience and sustainable development.

MAIN replicates the approach established and developed by Notre Dame University under the Notre Dame Global Adaptation Programme (ND-GAIN). Based on the ND-GAIN framework, a customised and localised MAIN framework was developed for the Malaysia State Index. The state index will assess the vulnerability and readiness level and produce the matrix and trends for Malaysia against climate change. The assessment result will leverage the public and private investment sectors and plan adaptive action for climate change possibilities. Through MAIN, the analyses and results are more accurate and collective than ND-GAIN since it uses data and information from Malaysia's government ministries and agencies.

Table 1 indicates the definitions of terminologies used in MAIN.

**Table 1.** Definitions of terminologies ([1]).

| Criteria | Definition |
| --- | --- |
| Vulnerability | Propensity or predisposition of human societies to be negatively impacted by climate hazards. |
| Exposure | The extent to which human society and its supporting sectors are stressed by the future changing climate conditions. Exposure in ND-GAIN captures the physical factors external to the system that contribute to vulnerability. |
| Sensitivity | The degree to which people and the sectors they depend upon are affected by climate-related perturbations. The factors increasing sensitivity include degree 4 of dependency on sectors that are climate sensitive and the proportion of populations sensitive to climate hazard due to factors such as topography and demography. |
| Adaptive Capacity | The ability of society and its supporting sectors to adjust to reduce potential damage and to respond to the negative consequences of climate events. In ND-GAIN, adaptive capacity indicators seek to capture a collection of means, readily deployable to deal with sector-specific climate change impacts. |
| Readiness | Readiness to make effective use of investments for adaptation actions thanks to a safe and efficient business environment. |
| Economic Readiness | The investment climate that facilitates mobilising capitals from the private sector. |
| Governance Readiness | The stability of the society and institutional arrangements that contribute to the investment risks. A stable country with high governance capacity reassures investors that the invested capitals could grow under the help of responsive public services and without significant interruption. |
| **Social Readiness** | Social conditions that help society to make efficient and equitable use of investment and yield more benefit from the investment. |

The index was calculated based on vulnerability and readiness scores on specific indicators significant for different sectors. The index matrix will enable stakeholders and

policy makers to plan adaptive capacity under the influence of climate change scenarios for global and local levels.

The following are the goals of MAIN:

a.    Improve state preparedness and reduce the vulnerability in various climate change sectors, such as food, water management, health, ecosystem, infrastructure, economic, governance and social.

b.    Public and private investment can be developed, executed, and managed strategically for physical and infrastructure projects.

c.    Revise and update National Climate Adaptation policy and plan, best management practices, standard operating procedures, technical guide and manual with comprehensive, collective, accurate, reliable, and up-to-date information.

The feasibility of MAIN offers the following outcome:

a.    Data and information—ability to provide comprehensive and collective information and data as input plan for adaptation and mitigation plan.

b.    Asset and resource management—improve efficiency and effectiveness of asset and resource management.

c.    Reduce risk and impact—ability to prevent, reduce and rescue high-risk areas from climate change impacts.

d.    Reduce loss and life—ability to identify and reduce losses (lives, properties, and ecosystem) in the event of disasters.

e.    Data management policy and governance—improve open data, data sharing, data quality and data retention initiatives of government ministries and agencies.

*2.2. MAIN Challenges and Issues*

The indicators used for MAIN were identified and selected based on experts' opinions through a series of workshops. The experts involved were selected based on their expertise, knowledge, and experience with climate change studies.

Through workshops, data gathering from experts allows more information extraction, diversified views and opinions, greater acceptability, degree of involvement, and encouragement of expert participation. Despite the advantages, the drawbacks of this approach are lack of responsibility, dominance, negotiation decisions and groupism, which will affect the decision quality. Therefore, a structured mechanism needs to be engaged in the process to avoid the issues for criteria identification and selection. On top of that, the selection of methodology used in the calculation also contributes to the MAIN development's complexity.

From a data perspective, the emergence of Big Data and data science will stimulate data to account for indicators selection. In MAIN, the volume, variety, and veracity of data have challenged the experts to identify, select and determine which data can be used as the indicators. Although the data volume is vital, prioritising and ranking the data as indicators will be based on availability, accessibility, and readiness. This scenario will encourage favouritism and conflicts of interest among the experts.

The issues with voluminous data have challenged the experts to identify and select which relevant and significant data to represent the sectors. The potential and possible data are enormous; thus, experts must assess and evaluate the data thoroughly to ensure the connotation of selected indicators with climate change vulnerability and readiness.

Data collection has challenged the experts to identify the correct data source and data owner of selected indicators. The data also must be available and accessible to be used. Data confidentiality, intellectual property and data security also need to be considered since government entities (ministries and agencies), non-government, private sectors, and government-link companies are involved, which might prolong the time for the data to be available.

In some cases, the collected data are for non-uniform periods, unstructured and incomplete, where they need to be transformed into a new dataset before being used. Hence, additional time is required for pre-processing and re-analysis of the data to become indicators. Therefore, extra time is consumed for the index to be fully generated.

### 2.3. Current MAIN Criteria

For the development of the MAIN, based on previous studies and current works, the experts were in consensus to use the final indicator as mentioned in Table 2 (indicator for vulnerability) and Table 3 (indicator for readiness).

**Table 2.** MAIN indicator for vulnerability sector.

| No. | Vulnerability Sector | Indicator/Criteria | Total |
|---|---|---|---|
| 1 | Water | a. Water 1—Exposure: Projected Change of Annual Water Yield<br>b. Water 1—Sensitivity: Water Stress Index<br>c. Water 1—Adaptive Capacity: Reserve Margin<br>d. Water 2—Exposure: Projected Change of Annual Low Flow<br>e. Water 2—Sensitivity: River Water Quality<br>f. Water 2—Adaptive Capacity: Supplementary Flow<br>g. Water 3—Exposure: Projected Change of Dry Spell in Irrigated Area<br>h. Water 3—Sensitivity: Number of Farmer Affected<br>i. Water 3—Adaptive Capacity: Dam and Pump Capacity<br>j. Water 4—Exposure: Projected Change in Evapotranspiration<br>k. Water 4—Sensitivity: Dam Intake<br>l. Water 4—Adaptive Capacity: Non-Revenue Water | 12 |
| 2 | Food & Commodity | a. Food and Commodity 1—Exposure: Projected Change in Palm Oil Yield<br>b. Food and Capacity 1—Sensitivity: Area Planted with Oil Palm<br>c. Food and Capacity 1—Adaptive Capacity: Ratio of Oil Palm Plantation to Total Oil Palm Planted Area<br>d. Food and Commodity 2—Exposure: Projected Change of Paddy Yields<br>e. Food and Commodity 2—Sensitivity: Paddy Planted Area<br>f. Food and Commodity 2—Adaptive Capacity: Ratio of Granary Area to Total Paddy Planted Area | 6 |
| 3 | Infrastructure | a. Infrastructure 1—Exposure: Projected Change of Flood<br>b. Infrastructure 1—Sensitivity: Number of People Affected<br>c. Infrastructure 1—Adaptive Capacity: Cost for Structural and Non-Structural Approaches for Flood Mitigation<br>d. Infrastructure 2—Exposure: Coastal Inundation Due to Sea Level Rise<br>e. Infrastructure 2—Sensitivity: Population Living Below 3 m Above Mean Sea Level<br>f. Infrastructure 2—Adaptive Capacity: Budget Spend on Structural and Non-Structural Coastal Protection<br>g. Infrastructure 3—Exposure: Projected Increase of Extreme Flow<br>h. Infrastructure 3—Sensitivity: Duration of Dam Overspill<br>i. Infrastructure 3—Adaptive Capacity: Funds Related to Dam Capacity Planning, Upgrades and Maintenance<br>j. Infrastructure 4—Exposure: Projected Change in Low Flow<br>k. Infrastructure 4—Sensitivity: Dependency on Mini Hydro for Energy Production<br>l. Infrastructure 4—Adaptive Capacity: Renewable Energy | 12 |

In contrast with ND-GAIN indicators, as shown in Tables 4 and 5, the chosen sector and number of indicators are different. Despite the sectors used for MAIN being three (3) sectors, (1) water, (2) food and commodity, and (3) infrastructure, the indicators used to represent the sectors are detailed and explicitly exclusive, which offer better information and analysis for the vulnerability index. In general, MAIN's vulnerability indicators focus more on modelling results, where additional analysis is required to analyse and produce the results. As for readiness, the sectors applied in MAIN are the same, but the type and number of indicators used for MAIN are more comprehensive, which can improve the trend analysis. Figure 1 compares the number of indicators for the vulnerability and readiness sector.

**Table 3.** MAIN indicator for readiness sector.

| No. | Readiness Sector | Indicator/Criteria | Total |
|---|---|---|---|
| 1 | Economy | a. Economy 1—Cost Increment in Adaptation Efforts<br>b. Economy 2—Business Opportunity and Continuity<br>c. Economy 3—Property Value and Productivity<br>d. Economy 4—Eco-Tourism<br>e. Economy 5—Alternative Economy for Local Residents<br>f. Economy 6—Carbon Dioxide Emission Intensity<br>g. Economy 7—Climate Change Impact Readiness to Water Infrastructure Cost<br>h. Economy 8—Economic Growth<br>i. Economy 9—Readiness to Pay Insurance for Disaster<br>j. Economy 10—Recovery Cost | 10 |
| 2 | Governance | a. Governance 1—Number of Civil Servants Working in Climate-Related Sector per 10,000 Population<br>b. Governance 2—Changes in Water Tariff<br>c. Governance 3—Number of Civil Servants Working in Disaster-Related Job Per 10,000 Population<br>d. Governance 4—Education<br>e. Governance 5—Provision of Access to Information and Communication Technology (ICT) Infrastructure | 5 |
| 3 | Social | a. Social 1—Number of Climate-Change-Related Programs within States<br>b. Social 2—Critical Flooding Facilities<br>c. Social 3—Reliability of ICT Infrastructure<br>d. Social 4—Quintile (20%) of Income of the State Population<br>e. Social 5—Percentages of Population with Tertiary Education | 5 |

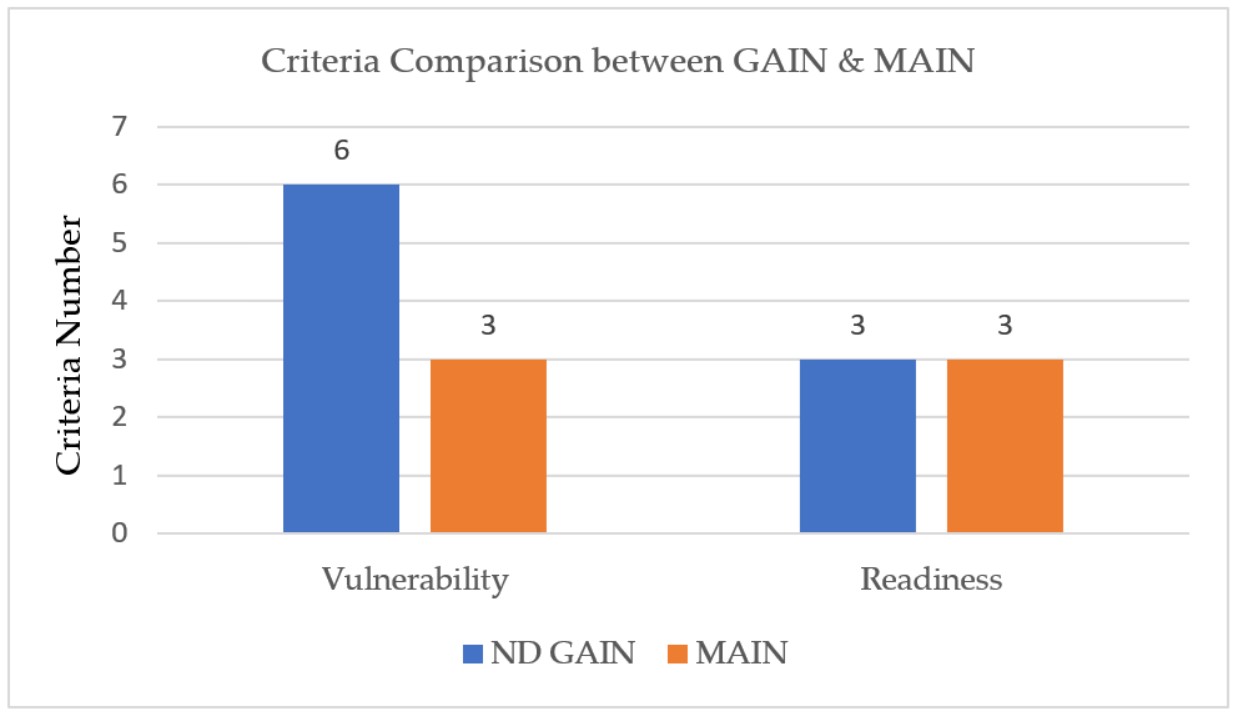

**Figure 1.** Criteria number of MAIN vs. ND-GAIN.

**Table 4.** Sector and indicator for vulnerability (ND-GAIN).

| No. | Vulnerability Sector | | Indicator/Criteria | Total |
|---|---|---|---|---|
| 1 | Food | a.<br>b.<br>c.<br>d.<br>e.<br>f. | Projected change of cereal yields<br>Food import dependency<br>Agriculture capacity (Fertiliser, Irrigation, Pesticide, Tractor use)<br>Projected population change<br>Rural Population<br>Child malnutrition | 6 |
| 2 | Water | a.<br>b.<br>c.<br>d.<br>e.<br>f. | Projected change of annual runoff<br>Freshwater withdrawal rate<br>Access to reliable drinking water<br>Projected change of annual groundwater recharge<br>Water dependency ratio<br>Dam capacity | 6 |
| 3 | Health | a.<br>b.<br>c.<br>d.<br>e.<br>f. | Projected change of deaths from climate-change-induced diseases<br>Slum population<br>Medical staff (physicians, nurses and midwives)<br>Projected change of length of transmission season of vector-borne diseases<br>Dependency on external resources for health services<br>Access to improved sanitation facilities | 6 |
| 4 | Ecosystem Services | a.<br>b.<br>c.<br>d.<br>e.<br>f. | Projected change of biome distribution<br>Dependency on natural capital<br>Protected biomes<br>Projected change of marine biodiversity<br>Ecological footprint<br>Engagement in international environmental conventions | 6 |
| 5 | Human Habitat | a.<br>b.<br>c.<br>d.<br>e.<br>f. | Projected change of warm period<br>Urban concentration<br>Quality of trade and transport-related infrastructure<br>Projected change of flood hazard<br>Age dependency ratio<br>Paved roads | 6 |
| 6 | Infrastructure | a.<br>b.<br>c.<br>d.<br>e.<br>f. | Projected change of hydropower generation capacity<br>Dependency on imported energy<br>Electricity access<br>Projection of sea level rise impacts<br>Population living under 5 m above sea level<br>Disaster preparedness | 6 |

**Table 5.** Sector and indicator for readiness (NG-GAIN).

| No. | Readiness Sector | | Indicator/Criteria | Total |
|---|---|---|---|---|
| 1 | Economic | a. | Doing business | 1 |
| 2 | Governance | a.<br>b.<br>c.<br>d. | Political stability and non-violence<br>Control of corruption<br>The rule of law<br>Regulatory quality | 4 |
| 3 | Social | a.<br>b.<br>c.<br>d. | Social inequality<br>ICT infrastructure<br>Education<br>Innovation | 4 |

### 3. Methodology

Based on a systematic literature review study conducted on an overview of the multi-criteria decision analysis (MCDA) application for managing water-related disaster events [2], the same literature was used to further analyse flood disaster events for this study. In total, 131 works in the literature, focused on flood disaster events, were further analysed based on the thematic analysis technique. The thematic analysis conducted covers two aspects for this study, which are (1) criteria identification and selection based on the PESTEL framework, and (2) flood measures and decision goals in the context of the disaster management plan (refer to Figure 2).

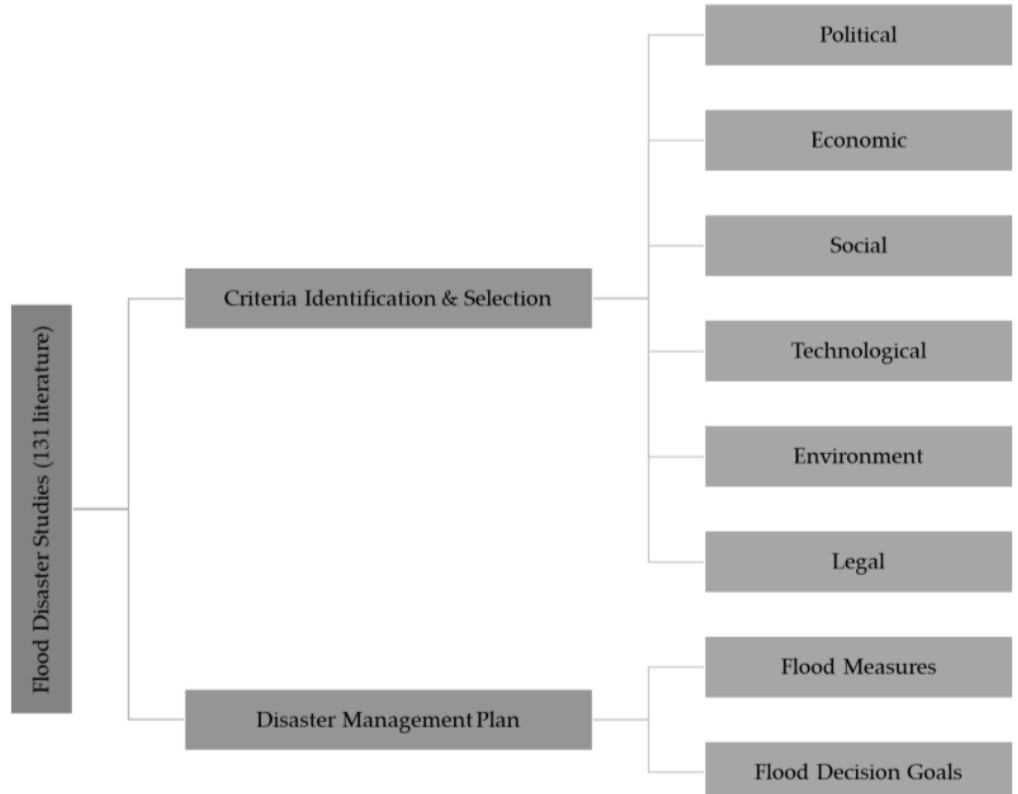

**Figure 2.** Thematic analysis structure.

Seven (7) metadata were extracted from the previous literature to be used in this study:

1. Author's name;
2. Article title;
3. MCDA technique;
4. DMP phase;
5. Criteria employed;
6. Type of flood measures;
7. Type of decision goals.

Descriptive statistical approaches were applied to undertake a detailed analysis of these metadata, while qualitative and descriptive methods were used for explanatory and discussion reasons. The graphs and tables that are supplied are for illustrative purposes only.

#### 3.1. Thematic Analysis

Further examination was conducted using the thematic analysis method to ascertain the criteria employed, flood measures, and decision goals metadata. For the criteria analysis, a thematic analysis was conducted using a six (6)-macro-domain-analysis strategic framework called PESTEL (political, economic, social, technological, environmental, and

legal). Meanwhile, the identified measures were mapped according to the most commonly used in the flood management plan for flood measures analysis. In the case of decision goals, the investigation was conducted by clustering the decision goals according to the objective for flood disaster management.

### 3.1.1. Criteria Analysis

The analysis strategic framework identifies significant macro domain criteria used for facilitating flood disaster management planning. This analysis helps to cluster the identified criteria into pertinent macro domains. The criteria specified in the previous studies were clustered into six (6) PESTEL domains. The phases involved are as follows:

a.      Phase 1: Remove duplication criteria.
b.      Phase 2: Create a theme for criteria.
c.      Phase 3: Identify distinct criteria.
d.      Phase 4: Clustering criteria according to PESTEL domain.

### 3.1.2. Flood Measures Analysis

Strategies adopted to deal with flood disasters are known as flood measures. It is the action taken in response to the problem analysis and a solution. The results of previous studies were used to determine the type of measures taken in the flood management plan in this study. The identified flood measures were mapped to three (3) major measures: (1) assessment, (2) mapping, and (3) assessment and mapping.

### 3.1.3. Decision Goal Analysis

The decision goals were analysed by clustering the goals based on the objectives of the flood measures from the previous studies. The decision goals were identified based on the literature's discussion's impacts, outcomes, or aims. There are five (5) distinct decision goals: (1) resilience, (2) risk, (3) hazards, (4) vulnerability, and (5) risk and resilience.

While the purpose of resilience is to reduce flood damage by learning to live with floods, the goal for hazards is to determine the likelihood of flood episodes that may cause harm. The goal for vulnerability in this context is to determine the ability and capacity to cope with flood susceptibility. The risk decision goal is to reduce or mitigate the negative effects and consequences of flooding.

## 4. Analysis and Results

### 4.1. Macro Domain (PESTEL Framework) Criteria Analysis

The PESTEL framework criteria analysis revealed a significant result in criteria selection for the flood management plan. The mapping of the criteria from previous studies according to this framework facilitate in (1) understanding which macro domain dominated the criteria selection; (2) improving criteria selection by identifying appropriate and relevant criteria; and (3) expanding the criteria selection through reviewing, assessing, and updating existing criteria based on the best practice applied in solving flood disaster events. Therefore, criteria analysis according to this framework allows more options of credible criteria that are available to be chosen and ready to be used for future flood management planning. This will act as a guideline for stakeholders and policy makers to decide which criteria should be selected.

According to previous studies, more attention was paid to criteria from a single macro domain than those from multiple macro domains. In comparison to other macro domains, the environment domain has received much interest. The distribution of literature across single and multiple macro domains is summarised in Table 6.

**Table 6.** Distribution of articles based on single and multiple domains.

| No. | Domain Group | Count |
|---|---|---|
| 1 | Single Macro domain | 73 |
| 2 | Integrated Macro domain | 58 |

While there were studies that concentrated on multiple macro domains, there was a severe lack of comprehensive criteria selection that encompassed all six (6) macro domains used in the previous study. Table 7 illustrates the distribution of criteria across macro domains based on the criteria used. Environment was the most highly focused macro domain relevant to the flood management plan's criteria. The majority of the environmental criteria were related to flooding analysis, including dataset for hydrology, hydraulic, water quality, land use, soil map, etc. Economic, social, and environmental criteria are the most integrated macro domains that have been studied. The five macro domains that are the most frequently combined are (1) economic + social + technological + environmental + legal, and (2) political + economic + social + technological + environmental + legal. Although studies have been conducted that combine criteria from multiple macro domains, future studies should consider encompassing all PESTEL macro domain criteria.

From the 131 analysed works in the literature, 1332 criteria were extracted and mapped using the PESTEL framework. Figure 3 illustrates the distribution of criteria according to PESTEL macro domains. Environmental, social, and economic were the top three (3) macro domains compared to other domains. This is both pertinent and rational, given that the main objective of the flood management plan is to lessen economic, social, and environmental impacts.

*4.2. Flood Measures Analysis*

Analysis of the flood measures indicated the most prominent flood measures applied from the previous studies. Having comprehensive flood measures based on previous studies help in future flood management planning in (a) understanding the type of flood measures commonly and suitably used; (b) replicating the flood measures strategies for the same characteristic of flood scenario; and (c) improving flood measure strategies (lesson learned from previous studies). Hence, analysis of the flood measures allows application and action of flood measures to be used and how it implicates data requirement and data collection. Three (3) flood measures identified from previous studies are (1) assessment, (2) mapping, and (3) assessment and mapping.

Assessment actions are used as non-structural measures to assess the feasibility and capability of planned action. This method can assist stakeholders and policy makers in developing more effective and efficient planning. The mapping measures enhances the visual representation of the location and actual state, assisting stakeholders and policy makers in disaster management planning. Additionally, it is prudent to consider the combination of the mapping and assessment measures. As a result of the preceding findings, it is obvious that combining measures is one way to improve the decision-making process.

Figure 4 depicts the flood measures taken in response to MCDA techniques for flood management. While the assessment measures dominate the measures used in flood management, the mapping measures are gaining traction as a growing action to assist in flood management plan analysis. The trend toward combining both assessment and mapping measures is becoming more prevalent, where both are used to visualise and analyse simultaneously.

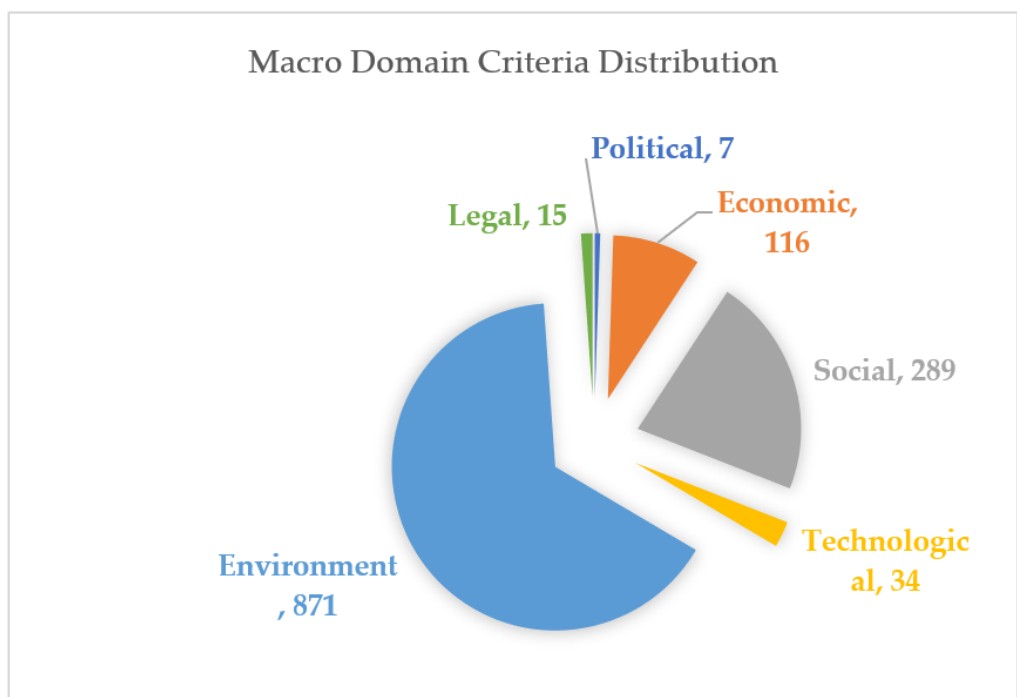

**Figure 3.** Distribution of criteria according to PESTEL domains.

**Table 7.** Distribution of articles based on PESTEL domains.

| No. | Macro Domain | Count |
|---|---|---|
| 1 | Environment | 68 |
| 2 | Economic + Social +Environment | 19 |
| 3 | Social + Environment | 11 |
| 4 | Economic + Social + Technological + Environment | 6 |
| 5 | Economic + Social + Technological + Environment + Legal | 4 |
| 6 | Political + Economic + Social + Technological + Environment | 3 |
| 7 | Social | 3 |
| 8 | Social + Technological + Environment | 3 |
| 9 | Economic | 2 |
| 10 | Economic + Environment | 2 |
| 11 | Economic + Social | 2 |
| 12 | Economic + Social +Environment + Legal | 1 |
| 13 | Economic + Technological + Environment | 1 |
| 14 | Economic + Technological + Legal | 1 |
| 15 | Political + Economic + Social + Environment | 1 |
| 16 | Political + Social + Economic | 1 |
| 17 | Social + Environment + Legal | 1 |
| 18 | Social + Technological | 1 |
| 19 | Technological + Environment | 1 |

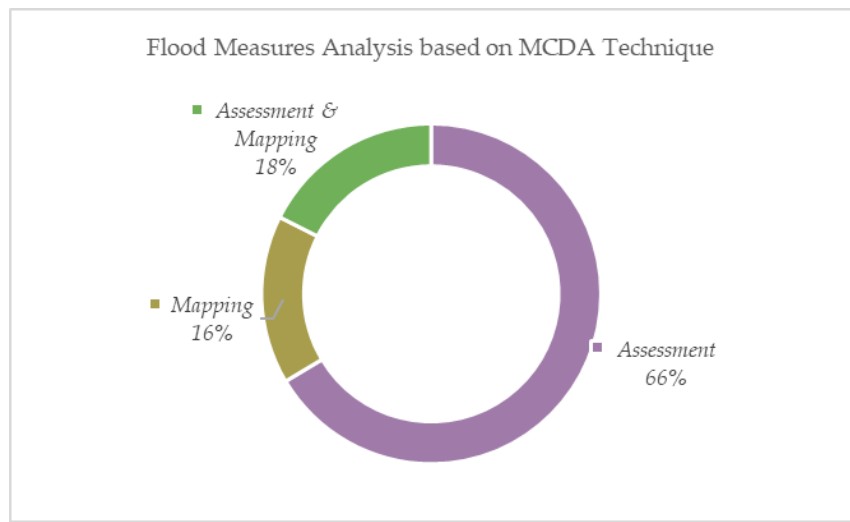

**Figure 4.** Flood measures analysis based on MCDA technique.

*4.3. Decision Goals Analysis*

Analysis on decision goals widening understandability on which goals are the main focus of stakeholders and policy makers. Classifying the decision goals for various flood measures aids in comprehending why these measures were taken. This information may assist stakeholders and policy makers in better understanding the decision goals that correspond to the flood measures that can be implemented in future flood management planning. In addition, the mapping can aid the exploration and diversification of decision goals in future management plans. The decision goals for each flood measure identified have been classified into four (4) categories: (1) resilience, (2) risk, (3) hazards, and (4) vulnerability.

Table 8 shows the clustering details for each flood measure according to decision goals. The table indicates that regardless of any flood measures action in the flood management plan, a study on flood vulnerability was the predominant source of concern. It could be influenced by the high importance of comprehending the possibility of return events.

**Table 8.** Flood measure and decision goals trends.

| Flood Measures | Decision Goals | No. of Articles | Percentage |
|---|---|---|---|
| Assessment | Resilience | 38 | 44% |
| | Vulnerability | 26 | 30% |
| | Risk | 17 | 20% |
| | Hazards | 4 | 5% |
| | Risk and Resilience | 2 | 2% |
| Mapping | Vulnerability | 13 | 62% |
| | Hazards | 5 | 24% |
| | Risk | 2 | 10% |
| | Resilience | 1 | 5% |
| Assessment and Mapping | Vulnerability | 14 | 61% |
| | Hazards | 3 | 13% |
| | Risk | 2 | 9% |
| | Resilience | 2 | 9% |
| | Vulnerability and Resilience | 1 | 4% |
| | Risk and Vulnerability | 1 | 4% |

Apart from vulnerability, goals for resilience and risk were emphasised as part of the strategy for managing flood disasters based on flood assessment. Meanwhile, the previous study's primary focus on mapping was comprehending the hazards and risks associated with flooding. Combining the two measures enables the achievement of additional goals for effective and efficient assessment decisions, such as vulnerability and resilience, and risk and vulnerability.

The criteria used to define each action and decision goal were clustered according to the PESTEL macro domain. Prior studies indicated that integrating criteria from other macro domains into flood management was considered. Despite the integration of macro domains, the macro domain selection was not exhaustive and did not encompass all domains in PESTEL. The five (5) prevalent domains were (economic + social + technological + environmental + legal). Meanwhile, studies focusing on a single macro domain explained that the primary focus was on the environment and social issues. Considering the flood disasters' impact more on the environment and society may have influenced the trend. Table 9 summarises the focus of decision goals considered under the assessment measure.

The decision goals for mapping demonstrate that criteria from a single macro domain, environment, was prioritised over those from the social domain to facilitate the creation of flood management maps. Maps were primarily used to identify flood-affected areas or locations. The visualised maps provide a better overview of locations for stakeholders and policy makers and aid in planning based on real conditions in the location. Most previous studies focused on determining the vulnerability level of a location, as shown in Table 10.

**Table 9.** Distribution macro domain based on assessment measures.

| Macro Domain (PESTEL) | Assessment | | | | |
|:---:|:---:|:---:|:---:|:---:|:---:|
| | Resilience | Risk | Hazards | Vulnerability | Risk and Resilience |
| Environment | [3–14] | [15–19] | [20–23] | [24–35] | [36] |
| Economic + Social + Environment | [37–41] | [42,43] | | [44–50] | |
| Economic + Environment | [51–53] | | | | |
| Economic + Social + Technological + Environment | [54–56] | [57] | | [58] | [59] |
| Economic + Social + Technological + Environment + Legal | [60–62] | | | [63] | |
| Economic + Social + Environment + Legal | [64,65] | [66] | | | |
| Social + Technological + Environment | [67,68] | | | | |
| Economic + Social | [69] | [70] | | | |
| Economic + Technological + Environment | [71] | | | | |
| Economic + Technological + Legal | [72] | | | | |
| Political + Economic + Social + Environment | | [73] | | | |
| Political + Economic + Social + Technological + Environment | [74] | [75,76] | | | |
| Political + Social + Environment | [77] | | | | |
| Social | [78] | | | [79] | |
| Social + Environment | | [80–83] | | [84–86] | |
| Social + Environment + Legal | | | | [87] | |
| Social + Technological | [88] | | | | |
| Technological + Environment | [89] | | | | |

**Table 10.** Distribution macro domain based on mapping measures.

| Macro Domain (PESTEL) | Mapping | | | |
|---|---|---|---|---|
| | Vulnerability | Resilience | Risk | Hazards |
| Environment | [90–100] | | [101] | [102–106] |
| Economic + Social + Environment | [107] | | | |
| Social | [108] | | | |
| Social + Technological + Environment | | | [109] | |
| Economic + Social | | [110] | | |

The majority of previous studies focused on determining the vulnerability level of a location. Incorporating other macro domains criteria was also considered in the flood management plan. The previous study utilised the integrated domains of (Economic + Social + Environment), (Social + Technology + Environment), and (Economic + Social). In this context, social becomes more desirable to use than other domains. No criteria from the political domain were used to determine flood measures in mapping. The challenges associated with identifying suitable data for the political domain may affect the selection. Data readiness and data format may also pose challenges in selecting criteria.

The possibilities are increased by combining two flood measures, assessment and mapping, as a hybrid approach to flood management. While the assessment measure provides a descriptive analysis on flood risks and recommendations for combating their impacts, mapping and visually representing the results may provide additional insight and improve understanding for a more effective flood management plan.

As illustrated in Table 11, the study focused on a single and specific macro domain dominated in previous studies. The macro domain environment dominated the criteria selection, followed by the social, economic, and technological domains.

**Table 11.** Distribution macro domain based on assessment and mapping measures.

| Macro Domain | Assessment and Mapping | | | | | |
|---|---|---|---|---|---|---|
| | Vulnerability | Risk | Hazards | Resilience | Vulnerability and Resilience | Risk and Vulnerability |
| Environment | [111–121] | [122] | [123–125] | [126,127] | [128] | |
| Economic + Social + Environment | [129] | | | | | |
| Economic + Social + Technological + Environment | | [130] | | | | |
| Social + Environment | [131,132] | | | | | [133] |

As demonstrated by previous studies, numerous factors may influence the criteria identification and selection (mapping macro domain criteria, flood measures, and decision goals). While big data may expand data possibilities, stakeholders and policy makers face challenges in selecting which data to employ. Data availability and readiness would drive the data selection from a data perspective. If data are deemed critical and vital for usage but are not available or ready, stakeholders and policy makers should continue to explore data substitution options.

Apart from that, the data type may affect the criteria selection. Infeasible and unquantifiable data types necessitate further data processing and analysis. Pre-processing data takes time; thus, leveraging appropriate data to represent the same insight can save time.

Future research should emphasise the significance of integrating multiple macro domain criteria and assessing the criteria from a PESTEL perspective considering these findings. There is a knowledge, method and application gap in flood management planning

that utilises decision analysis techniques and macro domain criteria. By closing these gaps, stakeholders and policy makers can develop a more strategic flood management plan.

The investigation of these findings reveals inconsistency in implementing flood mitigation measures and the wide variety of decision goals in the disaster management plan.

Numerous factors, including the following, may affect the scenario:

a.  Flood measures

    i.      Flood disaster history;
    ii.     Future flood risk area;
    iii.    Risk assessment;
    iv.    Data accessibility;
    v.     Data availability;
    vi.    Implementation capabilities.

b.  Diverse decision goals

    i.      Risk assessment based on geographical location, social and economic impacts;
    ii.     Future flood management plan;
    iii.    Increase flood measures effectiveness and efficiency based on the area's risk assessment.

### 4.4. Decision Analysis

The matrix table provided by [2] shows the possibilities of how decision analysis can be implemented specifically in criteria identification and selection for floor management plans (refer to Table 12). There are two (2) areas where MCDA can be applied, which are [a] criteria identification and selection for flood disaster management plan; and [b] improving existing MAIN criteria selection and identification.

**Table 12.** Finding on MCDA application for flood management [2].

| MCDA Technique | Flood | | | |
| --- | --- | --- | --- | --- |
| | **Mitigation** | **Preparedness** | **Recovery** | **Response** |
| AHP | [3,12,15,16,20–22,27,29,32,35,36,43,47,48,63,64,75,82,85–87,89,90,92,94,97,98,102,103,105,108,113,114,116,118–122,124,128,133] | [42,65,74,96,101,111,117,123] | [30] | [17,130,131,134] |
| Mixed methods | [5,10,11,24,33,44,54,56,61,66,70,73,79,83,84,99,100,109,112,129,132] | [14,23,28,34,37,39,57,59,77,127] | - | [60,78] |
| TOPSIS | [7,45,49,58,62,91,107,126] | [6,55,104] | - | [8,13] |
| ANP | [40,68,93,115] | [26,135] | - | [72] |
| CBD | [46] | - | - | - |
| CP | [9,25,38,52] | [51] | - | - |
| ELECTRE | - | - | - | - |
| Entropy | [106,125] | - | - | - |
| NAIADE | - | - | - | - |
| PROMETHEE | [41,53,67,69,71,80] | - | - | - |
| SAW/WSM | [19,76,81] | - | - | [4] |
| VIKOR | - | - | - | [50] |

In the context of criteria analysis from macro domain perspectives, the MCDA technique can identify and select which criteria to choose and prioritise within the same macro domain. The same technique can also be applied to choose and prioritise among macro domain PESTEL.

The application of structured decision analysis will close the gap and limitation faced by stakeholders and policy makers in criteria identification and selection within macro and between macro domains.

## 5. Discussion: Suggestions for Improvement

### 5.1. Criteria Selection for Flood Management Plan

It is critical to adopt acceptable, relevant, and pertinent criteria to current needs and situations based on assessment and understanding to assist stakeholders and policy makers in effectively planning and making flood decisions.

The data evolution offers numerous criteria identification and selection possibilities from various perspectives. Incorporating macro domain criteria can be beneficial for flood planning and management. The plan will be holistic and inclusive to accommodate requirements and interests from various entities. Thus, engaging the PESTEL framework in the criteria identification and selection process may aid stakeholders and policy makers in developing a broad understanding of relevant, widely used, and applicable criteria based on prior studies. It could be accomplished by replicating, revising, and improving existing criteria towards the decision goals. As a result, stakeholders and policy makers would have a layout of options that can be referred to facilitate criteria identification and selection.

The following actions can be taken to refine the criteria selection process within the MAIN application:

a. Criteria selection based on the relative importance of each sector;
b. Incorporating PESTEL criteria into each sector to establish collective and comprehensive criteria options;
c. Incorporating macro domain criteria in each phase of DMP based on the MAIN index for managing water-related disasters;
d. Incorporating MAIN index and PESTEL criteria for decision goals in the flood management plan.

### 5.2. Flood Management Plan

Based on the index and existing criteria from MAIN, an improvement in criteria identification from the macro domain could be applied in strategising and planning a holistic DMP. Engaging the PESTEL and MAIN criteria in each phase of the DMP, the following benefit would be realised:

a. Determine location at risk which requires additional and immediate attention for flood management plan based on MAIN's index vulnerability and readiness;
b. Determine which additional criteria could be incorporated with the existing MAIN criteria to improve assessment;
c. Increasing criteria number on each sector to improve flood management plan;
d. Determine DMP's importance phase based on MAIN assessment.

### 5.3. Use of MCDA Method

The current approach used in MAIN's criteria identification and the selection seems to have drawbacks, as discussed in Section 2.2. Thus, the MCDA approach would be relevant to be introduced in the process, supporting the input and basis for decision making. Previous studies have shown how prevalently the technique has been applied in managing flood disaster management in various phases of DMP. Its application has gained traction, regardless of whether it is employed alone or with other methods. The mixed-method technique could be applied due to its benefits, correcting the single method deficiencies, and improving outcomes.

The potential use of the MCDA technique could be explored and applied in MAIN. This technique would assist not only stakeholders and policy makers, but also technical and non-technical expertise in making more structured, informed, and effective decisions.

### 5.4. Proposed Framework: MCDA Application for Flood Disaster Management Based on PESTEL Framework

The findings from this study from the context of the decision analysis technique, the macro domain criteria, and BD's impact have resulted in developing a framework for the future flood management plan. The proposed framework could guide stakeholders and policy makers to close the gaps from current limitations and challenges. Additionally, it could serve as a roadmap for MAIN initiatives to optimise and expand its application.

By incorporating structural, non-structural, and mixture flood measures, the proposed framework shown in Figure 5 enhances the DMP for all phases. Regardless of the measures taken, the framework recommends performing a macro domain analysis in which all possible criteria from the respective PESTEL domains must be considered and explored. The criteria for identification and selection will be determined by their significance and relevance to the DMP being implemented, which will be analysed using the MCDA technique. Stakeholders and policy makers will employ the MCDA technique to select, rank, sort, and/or describe the criteria used. The data and/or information for the selected criteria will be grouped based on historical and projected data. Historical data will include observed or measured data from previous events, such as rainfall trends, temperature trends, and flood events. In comparison, projection data are a collection of forecasting data that have been processed to create a future dataset, for example, a 100-year rainfall projection, projected population, and so forth. The required data will be further classified into three (3) data types: (a) structured, (b) unstructured, and (c) hybrid. Understanding the different data formats for the various data types will aid in implementing flood measures.

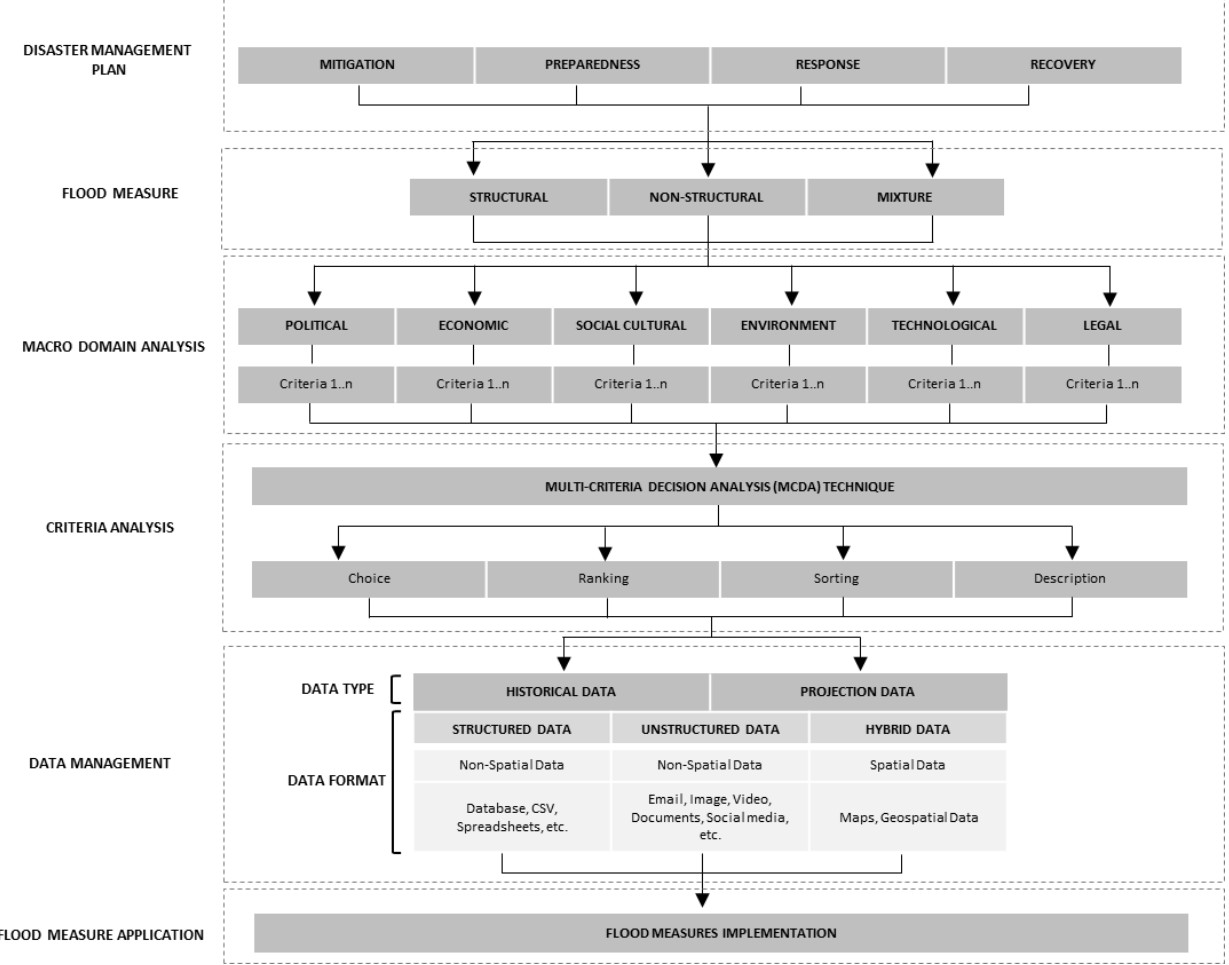

**Figure 5.** Proposed MCDA method framework for flood disaster management based on macro domain parameters PESTEL analysis.

According to the proposed framework in Figure 5, the number of criteria for each macro domain is expected to vary. Thus, to assist stakeholders and policy makers in identifying the most critical criteria, decision analysis techniques could be applied within the macro domain analysis. Through this, all criteria will be weighted by experts, ensuring that no criteria are overlooked, dominated, or influenced by experts' interests and preferences, among other things. Policy makers and stakeholders will benefit from the proposed additional step in the following areas:

a. Identifying the macro domain criteria which are mostly influenced and impactful;
b. Identifying and choosing the criteria within each macro domain criteria that highly impact and influence;
c. Identifying and prioritising the macro domain criteria that have the highest impact and influence.

*5.5. Potential Future Research*

According to this study's analysis and discussion, there are potential future research opportunities to support the implementation of MAIN initiatives while also optimising and diversifying the use of MAIN to address the effects and impacts of climate change, particularly in water-related disasters.

The following are some possible areas of future research:

a. A review of the current MAIN criteria used to calculate the vulnerability, readiness scores and adaptation index. The aim is to improve the criteria selection by incorporating macro domain criteria for an inclusive decision.
b. Develop flood forecast maps for high-risk locations based on the MAIN index, improve and revise criteria. The aim is to facilitate more thorough and collaborative decision making.
c. Bank criteria—a data repository of criteria (current and possible) that potentially can be used in flood management planning. The aim is to ensure data readiness and availability in the MAIN assessment.

**6. Conclusions**

The findings from this study have introduced a significant approach to the project management team of MAIN to improve its process, especially in criteria components and how it can lead to the improvement in flood management planning based on the MAIN index.

Expanding the current MAIN criteria based on the PESTEL analysis framework would assist in obtaining understanding from the overall perspective of macro domains. Criteria based on the PESTEL framework would provide broad and significant information to stakeholders and policy makers in flood management planning by considering the flood measures taken and their implication on the decision goals. These factors would have a significant impact on the overall DMP in every phase.

In addition, the introduction of the MCDA technique would improve the criteria identification and selection with a more structured and cohesive approach, compared to the current MAIN approach. Improvement in this process will ensure that identified criteria chosen by experts will be weighted and assessed to determine their relevancy, importance, and rank. Thus, it will lead to a more effective and efficient index.

From the overall analysis and findings, this study has proposed a framework that could be used as a guideline in implementing flood management plans based on the MAIN index, data evolution scenario, flood measures and criteria selection process in MAIN. The framework can be replicated to manage other disasters, such as droughts. It can be further improved to cater to issues and challenges in disaster management, data management and processing, and the disaster application strategy.

**Author Contributions:** Conceptualisation, M.F.A.; methodology, M.F.A.; validation, M.F.A., Z.Z., S.Y.T., N.H.A.G., A.M.J. and M.Z.M.A.; formal analysis, M.F.A.; investigation, M.F.A. and Z.Z.; resources, M.F.A. and Z.Z.; data curation, M.F.A.; writing—original draft preparation, M.F.A. and Z.Z.; writing—review and editing, S.Y.T., N.H.A.G., A.M.J., M.Z.M.A. and N.A.M.; visualisation, M.F.A. and N.A.M.; supervision, N.H.A.G., A.M.J. and M.Z.M.A.; project administration, A.M.J. and N.A.M. All authors have read and agreed to the published version of the manuscript.

**Funding:** This research received fund from National Water Research Institute of Malaysia (NAHRIM).

**Institutional Review Board Statement:** Not applicable.

**Informed Consent Statement:** Not applicable.

**Data Availability Statement:** The data presented in this study are available on request from the corresponding author. The data are not publicly available due to ongoing study.

**Acknowledgments:** This work was supported by National Water Research Institute of Malaysia (NAHRIM). Thank you to the researchers and management of NAHRIM for their useful input.

**Conflicts of Interest:** The authors declare no conflict of interest.

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
