# Peer review of "Big Data in Criteria Selection and Identification in Managing Flood Disaster Events Based on Macro Domain PESTEL Analysis: Case Study of Malaysia Adaptation Index"

_2504-2289, doi:10.3390/bdcc6010025_

Round 1

Reviewer 1 Report

This study performs a systematic literature review and thematic analysis using the PESTEL (Political, Economic, Social, Technological, Environment and Legal) framework. 

The paper investigates criteria identification and selection from a macro domain perspective in order to improve flood management planning.

The paper is interesting and well structured. My only concerns are the following:

- Literature analysis: I find that the authors performed an extensive review. The only lacking domain is that of renewable energy. I think that the study would benefit from the addition of relevant studies in this domain (see DOI: 10.1145/2790755.2790762, 10.3390/en13112907, 10.1109/ACCESS.2020.3019095, 10.3390/en15031082).

- Figure 2: Please check the size and make sure it is not in overfull.
Moreover, the box at the top has text "131 Literature". What does 131 mean?

Overall, the findings from this study can be exploited to achieve improvements in flood management planning based on the MAIN index. I suggest the authors to provide an improved version of the paper.

Author Response

The authors appreciate the comments and suggestion given by the reviewer to improve the paper. We are glad that the objective of the proposed framework would benefit future research in flood management planning. Below are the authors’ responses based on reviewer comments.

A.      Suggested new domain (Renewable Energy)

For the renewable energy domain, the authors believed the domain could be categorised under Environment based on macro domain PESTEL analysis framework. Thus, any criteria relevant to renewable energy will be classified under the Environment domain. Based on our work, criteria under renewable energy has been discussed from an infrastructure perspective. However, other criteria such as billing, availability, production, method and others could be considered as well in the future as part of the criteria to be assessed.

B.       Figure 2 – size

The size and layout of the figure have been resized to fit with the template.

C.       Figure 2 – 131 literature

The authors have changed the text “131 literature” to “Flood Disaster Studies (131 literature)” to improve readers’ understandability.

Reviewer 2 Report

The paper  presents a  study  that goal to explore and identify potential and possible criteria to be incorporated in the current flood management plan and identify the type of measures and decision goals necessary to facilitate flood management planning decisions
In this study, the authors have used seven metadata were extracted from the specialty literature.
The authors have proposed a framework that can be used as guidance in dealing with and optimizing the criteria based on challenges and the current application of Big Data and criteria in managing disaster events. The proposed framework has a wide utility so it can be the basis of other research and can also be used to manage other disasters. 

Author Response

The authors appreciate the comments given by the reviewer. We are glad that the proposed framework could be used as a reference and guide in future research specifically for inter-discipline research.

We also hope the proposed framework would benefit the criteria identification and selection for various disaster management in the context of a big data environment.

Reviewer 3 Report

Climate changes deeply influence environmental disasters such as landshifts, avalanches, floods and other. Therefore disaster management procedures are of great relevance for a country administration.

This paper deals with flood disaster management policies in Malaysia by comprehensively reviewing and classifying criteria found in the current literature.  However several remarks are in order:

  • in the title, emphasis in given on Big Data Analytics (BDA), but actually it only motivates the study -  "... Voluminous, veracity and var-72 sity data are among the positive impacts gained through BDA, allowing more analytical 73 processes to be conducted...." -  while no such procedures are examined in the paper. Therefore I suggest to change the title removing "Impact of Big Data Analytics in".
  • The abstract is very long and it looks like part of the conclusions; it should be more concise, focusing on what the reader should expect in the paper.
  • The indicator/criteria in Table 1 (e.g. Exposure. sensitivty, ...), and  in general each property (e.g.  Resilience, Risk, Hazards,   Vulnerability, ...), should be clearly defined in advance.
  • there are so many acronyms in the paper that, after a while, the reader gets lost in their meaning. Try to reduce the use of acronyms at least for short names/phrases.
  • there are several errors and language flaws (e.g. comprehending ---> understanding, ...) which must be fixed.

-

Author Response

The authors appreciate the comments and suggestions given by the reviewer to improve the paper. Below are the authors’ responses based on the reviewer’s comments.

A.      Title

The authors agreed to change the title by removing the word “Analytics” as the whole paper is more related to big data concepts compared to the big data analytics process. Thus, the final title for this paper is “Big Data in Criteria Selection and Identification in Managing Flood Disaster Events based on Macro Domain PESTEL Analysis: Case Study of Malaysia Adaptation Index.

B.       Abstract

The authors would like to retain the abstract of this paper because we had summarised each section of this paper so that the readers could understand the flow of this study. We would appreciate it if the reviewer could consider accepting the abstract as per the original version.

C.       Definition

The definition for terms and terminologies used in this paper have been added in Table 1 and lines 258 to 262.

D.      Acronym

The authors have reduced the number of acronyms in this paper. For example, in Table 2 and Table 3.

E.       Language

The authors have made a correction on language errors and mistakes without changing the meaning of the phrases.